# Examining the Psychoactive Differences between Kratom Strains

**DOI:** 10.3390/ijerph20146425

**Published:** 2023-07-21

**Authors:** Guido Huisman, Maximilian Menke, Oliver Grundmann, Rudy Schreiber, Natasha Mason

**Affiliations:** 1Department of Medicinal Chemistry, College of Pharmacy, University of Florida, Gainesville, FL 32611, USA; huisman.g@ufl.edu (G.H.); grundman@ufl.edu (O.G.); 2Department of Neuropsychology and Psychopharmacology, Faculty of Psychology and Neuroscience, Maastricht University, 6229 ER Maastricht, The Netherlands; m.menke@student.maastrichtuniversity.nl (M.M.); rudy.schreiber@maastrichtuniversity.nl (R.S.)

**Keywords:** kratom, *Mitragyna speciosa*, psychoactive compound, mitragynine, 7-hydroxymitragynine

## Abstract

Kratom (*Mitragyna speciosa*) is a Southeast Asian plant containing various alkaloids that induce pharmacological effects in humans. In Western countries, online vendors sell a variety of different kratom strains which are marketed to have distinct effect profiles. However, as of yet such marketing claims are unsubstantiated, and therefore the current study investigated whether differently colored kratom products can induce distinct effects, as self-reported by users. Six hundred forty-four current kratom users were anonymously surveyed to compare the self-reported effects of and motivations for using kratom products sold as red, green, and white strains. Most of the survey respondents were customers of the same kratom vendor, the products of which had been analyzed for their alkaloid content by an independent laboratory. The survey respondents reported distinct subjective experiences for different kratom strains, in a manner congruent with common marketing descriptions. However, the product analyses revealed no significant cross-strain differences in alkaloid content, suggesting that the reported effect differences might be disproportionally influenced by marketing narratives and anecdotal reports. Future studies should engage a more diverse population and include kratom strains from various vendors. Controlled, blinded experiments could assess whether the reported effect differences stem from a placebo effect or from alternative factors, e.g., minor alkaloids and terpenes.

## 1. Introduction

*Mitragyna speciosa*, colloquially known as kratom, is a tree that grows natively in several Southeast Asian countries, including Indonesia, Malaysia, and Thailand. In these countries, the human consumption of kratom dates back several centuries and has had medicinal and recreational motives [1,2]. Locals ingest the crude plant products either by directly chewing on the leaves, or by using the leaves as an ingredient in tea and other drink concoctions [3]. Through these routes of administration, the plant’s analgesic properties have been utilized to combat chronic pain, whereas its energizing effects have been popular among farming communities where kratom is used to prolong physical labor [4]. There are also records describing the use of kratom during religious ceremonies [5]. Although several parts of the tree have been analyzed for their phytochemical composition, only the leaves of Mitragyna speciosa are used for medicinal and recreational purposes [6]. However, in its native Southeast Asia, no distinction is made between kratom strains of different colors, nor are they advertised to be associated with distinct effects [7].

Interestingly, some of the different effects associated with kratom use appear to be dose-dependent, with low to moderate doses (1–5 g) inducing stimulation and awareness, and with moderate to high doses inducing analgesia and sedation [8]. While much of kratom’s pharmacology remains unexplored, mitragynine is widely regarded as one of kratom’s essential psychoactive ingredients. This indole alkaloid is generally found to be the most abundant alkaloid in kratom, accounting for about ⅔ of the plant’s alkaloid composition [2]. Other noteworthy kratom alkaloids resemble mitragynine, with some being similar in their molecular formulas and with others being formulaically identical but geometrically rearranged. Examples include mitragynine’s oxidized derivative 7-hydroxymitragynine and several mitragynine diastereomers including speciogynine, speciociliatine and mitraciliatine [9].

Mitragynine has been studied most extensively in recent years by means of animal models and in vitro studies The indole alkaloid has been shown to exert activity at the µ-opioid receptor as a partial agonist, and at the α_2_ adrenergic receptor as an agonist [10]. Synergistic activity at both receptors has been shown to exhibit antinociceptive effects in animals. The in vivo metabolite of mitragynine, 7-hydroxymitragynine, is a specific and selective partial agonist at the µ-opioid receptor associated with analgesic effects. The minor alkaloids paynantheine and speciogynine have been shown to agonize serotonin receptors, specifically 5-HT_1A_ and 5-HT_2B_ receptors, in vitro and in animal models [11]. This activity has been associated with potential mood-enhancing effects as observed in behavioral animal models of depression and anxiety [12,13].

In the West, kratom is marketed and sold as different strains, which are generally named after two properties: first, the leaf vein coloring of the plant products (e.g., red, green, or white) and second, the country or region the plant was harvested (Malaysia, Sumatra, Thailand, etc.). This gives rise to strain names such as red Malay, white Thai, and green Thai. Importantly, the marketing of kratom strains posits that different strains can produce distinct (and sometimes contradictory) pharmacological effects. For example, online vendors report that the kratom strain “Maeng-Da” (originating from Thailand), is an excellent energy booster and mood enhancer, whereas the kratom strain “Sumatra” (originating from Indonesia), is a good stress reliever [14]. Similarly, with regards to the different color denominations, the marketing of kratom products and anecdotal reports from kratom users, both commonly state that red kratom strains tend to be anxiolytic and calming whereas white and green strains tend to be stimulating and energizing. Representative effect descriptions of red, green, and white kratom strains are summarized in Table 1.

Presumably, these reported differences between kratom strains could be mediated by variations in their alkaloid profiles, in a manner analogous to cannabis strains producing distinct effects due to variations in cannabinoid profiles [15]. Indeed, findings from analytical chemistry suggest that different strains of kratom can vary in their alkaloid content [16]. Moreover, one study analyzed the elemental ingredients of kratom samples (e.g., calcium) by means of discriminant function analysis, and found that the samples could be reliably classified according to origin, sub-origin, and strain [17]. While such findings support the notion that the ingredients of different kratom products can vary substantially, thus far no empirical research has explored whether kratom products sold as different strains are reliably associated with distinct effect profiles. One phytochemical study did examine the metabolomic profile of young and mature kratom leaves and identified five unique alkaloids between the young and mature leaves while 76 secondary metabolites were present in both leaf samples, albeit in different concentrations [18]. While the major alkaloid mitragynine is present in approximately equal amounts in both young and mature leaves, some of the other indole alkaloids, including speciogynine, speciociliatine, and 7-hyroxymitragynine, are present in higher abundance in mature leaves. In most kratom leaf materials available on the US market, mature leaves are used.

Another study examined the seasonal and geographical differences between red- and green-veined kratom in parts of Thailand [19]. Though mitragynine remained the most abundant alkaloid, the total alkaloid concentration was much lower in the fall (October) sample compared to winter (January) and summer (June) samples. Regional differences indicate that total alkaloid content does also vary. Despite these phytochemical differences in composition, it is not clear whether the reported effect differences between kratom strains might just reflect a placebo or expectancy effect elicited by marketing, customer reviews or hear-say [20].

Therefore, the current study strived to investigate the notion that kratom products marketed as red, green, and white kratom strains can produce distinct pharmacological effects in humans. This was investigated by means of an online questionnaire which asked respondents about their motivations to use different kratom strains and the subjective effects they experienced when consuming them. It is important to note that most of the participants of this study were customers of the same kratom web shop (Super Speciosa, Super Organics LLC, St. Petersburg, FL, USA), which means that the results produced by this study might have limited generalizability to kratom products of other vendors. However, in addition to the survey results, a second source of data utilized in the current study was Certificates of Analyses (COAs) showing the alkaloid content of the kratom products consumed by those who indicated using kratom from the specific vendor surveyed in this study. These data were obtained through an independent laboratory unaffiliated with Super Speciosa, and therefore, the COAs allowed for the investigation of potential correlations between the alkaloid content of different kratom strains and the self-reported subjective effects produced by those strains.

## 2. Materials and Methods

### 2.1. Power Calculation

An online sample size calculator (https://www.surveymonkey.com/mp/sample-size-calculator/, accessed on 17 July 2023) was used to estimate the required number of survey respondents. Given that the hypotheses of the current study were novel and given that the main outcome measure was designed for this study specifically, the sample size calculation could not be based on prior research findings. However, the calculation assumed that the overall kratom use population is about 5 million people in the U.S., which is a conservative estimate based on kratom imports from Indonesia [21]. Given this assumption and given a 95% percent confidence interval, the required sample size was estimated to be 385. However, since not all kratom users consume all strains of kratom, the required sample size per strain was estimated to be about 150 responses. Under the assumption that 70% of responses were to be valid responses, it was estimated that the current study would require 315 valid responses (105 responses per strain) to have sufficient statistical power.

### 2.2. Participants

The target population of this survey were people who use kratom regularly. Recruitment of study participants was achieved in collaboration with the kratom vendor Super Speciosa (https://superspeciosa.com/, accessed on 17 July 2023). The survey URL was distributed as part of newsletters sent to clientele of Super Speciosa. No incentives of any kind were given for study participation. Survey responses were obtained between 22 July 2022 and 12 September 2022, after which a total of 644 responses were obtained. Informed consent was obtained at the beginning of the survey from all participants (see Appendix A). Ethical approval was obtained from the Ethics Review Committee of Psychology and Neuroscience of Maastricht University (ERCPN-226_101_08_2020_A2).

### 2.3. Procedure

The study was conducted using the online survey platform Qualtrics (Qualtrics, Provo, UT, USA). The survey consisted of a questionnaire which asked participants about their experiences with different kratom strains. The questionnaire was designed to be completed in approximately 5–15 min. On the first page of the survey, participants were provided with an informed consent form explaining the purpose of the study and the risks and benefits of participation. Participants were required to indicate their agreement to participate in the study by checking a box on the informed consent page. Participants were informed that withdrawing from the study was possible at any time and that the survey data could not be traced back to any individual.

### 2.4. Questionnaire Design

The questionnaire consisted of different blocks which are summarized below. A Qualtrics export of the full survey can be found in Appendix A.

#### 2.4.1. Block 1: General Health

Participants were asked to indicate their height (in ft. and inches) on visual sliders and were asked in open-ended format what their body weight was (in lbs). Subsequently, participants were asked in multiple-choice format about the frequency with which they smoke (i.e., cigarettes or nicotine-containing e-liquids) and the frequency of their alcohol consumption. Participants were then asked if they were currently prescribed antidepressants, anxiety medication, antipsychotics, opioids and/or other prescription medications. Lastly, respondents were asked whether a physician had ever diagnosed them with any of the following conditions: post-traumatic stress disorder (PTSD), depression, social anxiety, generalized anxiety disorder, schizophrenia (or other psychotic disorders), bipolar disorder, personality disorder, attention deficit or hyperactivity disorder (ADD/ADHD), addiction and substance disorder, fibromyalgia, rheumatoid arthritis, chronic pain. These specific diagnoses and medical prescriptions were included as items because prior literature suggested they are commonly relevant in the population of kratom users [22].

#### 2.4.2. Block 2: Kratom Strain Color

This block consisted solely of the following multiple-choice question: “Which color(s)/strain(s) of kratom do you generally consume?”. This question aimed to ascertain if the kratom products consumed by the respondent were green, white and/or red kratom strain products. The subsequent blocks of questions were repeated once, twice, or thrice, depending on whether the respondent consumed one (e.g., only red), two (e.g., red and green) or three (red, green, and white) different kratom strains (Figure 1).

#### 2.4.3. Block 3: Kratom Dosing Regimen

Block 3 of the survey was repeated for each kratom strain consumed by the respondent. Respondents were asked whether they purchased the given strain of kratom at Super Speciosa or at a different vendor. Respondents were asked what type of formulations (power, tablets, capsules and/or tea) they purchased of the given strain. On visual sliders (ranging from 0 to 20, with the items being gram, capsules, tablets, tea bags, teaspoons of powder, and tablespoons of powder), the respondents then indicated what serving size they typically consume of the given strain of kratom, at what frequency per week/month they consume the strain and what number of servings they consume in a typical day. Lastly, respondents were asked at what time of day (morning, afternoon, evening or at night) they generally consume the given strain of kratom, and whether they consume their kratom before, after or with a meal.

#### 2.4.4. Block 4: Ranking Motivations to Use Kratom

Block 4 of the survey was repeated for each kratom strain consumed by the respondent. The respondents were given a list of 13 prewritten statements indicating potential motivations to consume kratom. The respondents were asked to rank these motivations in the order most applicable to their consumption of the given strain of kratom (i.e., red, white, or green kratom products). This questionnaire was designed as an extended version of the 18-item reasons for drug-use scale by Boys and colleagues [23]. The added items included motives related to social context, self-exploration, and escapism, as derived from qualitative interviews [24]. The full list of motivational statements can be found in Appendix A.

#### 2.4.5. Block 5: Self-Reported Effects of Different Kratom Strains

Block 5 of the survey was repeated for each kratom strain consumed by the respondent. Respondents were shown a randomized list of items describing subjective drug effects (e.g., feeling happier), and were asked to indicate on a visual analog scale (VAS) ranging 0–100 to what extent they experienced the given drug effect. The meaning of the range of values on the VAS was exemplified with the following 4 labels: ‘The effect is not present or not applicable to my situation’ (i.e., 0), ‘The effect is present to some degree’ (i.e., 1–25, ‘The effect is clearly present’ (i.e., 26–75), and ‘The effect is present with great intensity’ (i.e., 76–100). The full list of items is shown in Appendix A.

### 2.5. Data Analysis

The survey data were analyzed using descriptive statistics and inferential statistics. Descriptive statistics were used to summarize the demographic characteristics of the participants and the reported effects of each kratom strain. Inferential statistics were used to test for differences in reported effects between different kratom strains. Correlational analysis of nominal data was conducted using a Chi-square or Friedman test while Student’s t-test and one-way ANOVA analysis followed by post-hoc Bonferroni comparison were used for interval data. Statistical significance is defined as α ≤ 0.05. Statistical analyses were conducted using SPSS software (version 26, IBM, Armonk, NY, USA).

## 3. Results

### 3.1. General Demographic Information

Table 2 summarizes the general demographic information of the survey population. The survey population comprised mostly middle-aged adults, with most respondents being within the age range of 35–44 years (29.70%), followed by 25–34 years (20.00%), and 45–54 years (22.80%). In terms of gender, the sample was relatively balanced, with slightly more respondents identifying as male (56.60%) compared to female (41.10%), and a small percentage identifying as non-binary (1.50%).

Most participants had received at least some college education (40.10%) or a bachelor’s degree (24.80%). Most participants were employed for wages (64.00%), with a sizeable percentage being retired (11.40%) or unable to work (8.90%). Nearly half of the respondents were married (44.50%), while one third had never been married (32.10%). In terms of ethnicity and nationality, the survey population was overwhelmingly Caucasian (86.40%) and American (85.98%), with limited representation of other ethnicities and nationalities. The annual household income distribution was varied, with the largest proportion of participants reporting an income of $100,000–$149,999 (16.00%), and with some respondents preferring not to disclose their income (7.50%). Most respondents reported never or rarely smoking and/or vaping (62.70%), while a considerable percentage reported daily use (32.80%). Most respondents rarely or never consumed alcohol (76.10%).

### 3.2. Clinical Profile

Table 3 displays the clinical profile and medication use of the survey population. In this sample, 29.0% used antidepressants, 14.3% anxiety medications, 3.3% antipsychotics, 7.1% opioid pain killers, and 9.3% stimulants. 23.8% of respondents reported using other prescription medications, while 44.6% were not taking any prescription medications. The most prevalent clinical diagnoses were depression (42.7%), generalized anxiety disorder (32.8%), and chronic pain (38.8%). Less prevalent diagnoses included PTSD (20.8%), social anxiety disorder (17.9%), ADD/ADHD (19.9%), addiction/substance use disorder (14.3%), fibromyalgia (8.7%), and rheumatoid arthritis (7.9%). Schizophrenia, bipolar disorder, and personality disorder were reported by 1.6%, 8.7%, and 3.0% of participants, respectively, while 22.0% reported not being diagnosed with any of the listed diagnoses.

### 3.3. Between-Strain Comparison of Motivations

Respondents were shown 13 randomly ordered motivational statements and were asked to rank these statements in the order most applicable to their use of a given kratom strain. Table 4 shows the average positions (between 1 and 13) at which each motivational statement was ranked for respondents consuming red (N = 184), green (N = 288), or white (N = 131). A Chi-squared test was conducted for each motivational statement to detect between-strain differences in the average ranking position.

Significant differences were found for the statements “to treat a medical condition” (*p* = 0.049), “to help you relax or sleep” (*p* ≤ 0.001), “to improve your mood or to feel less sadness/depression” (*p* = 0.007), “to help you concentrate work or study” (*p* = 0.001), “to induce or enhance a spiritual experience” (*p* = 0.021), and “to be more sociable or to get more enjoyment out of social events” (*p* = 0.005).

For the statement “to treat a medical condition”, red kratom users ranked it highest (position 3.51, SD = 3.37), while green strain users ranked it lowest (position 4.25, SD = 3.51). White kratom users ranked the statement at position 4.18 (SD = 3.38). For the statement “to help you relax or sleep”, red kratom users ranked it highest (position 3.83, SD = 2.54), while white strain users ranked it lowest (position 6.06, SD = 2.99). Green kratom users ranked the statement at position 5.15 (SD = 2.68). For the statement “to improve mood or feel less sadness/depression”, green kratom users ranked it highest (position 5.28, SD = 4.00), while red strain users ranked it lowest (position 6.34, SD = 3.88). White kratom users ranked the statement at position 5.45 (SD = 3.96). For the statement “to help concentrate on work or study”, white kratom users ranked it highest (position 5.30, SD = 3.49), while red strain users ranked it lowest (position 6.97, SD = 3.15). Green kratom users ranked the statement at position 5.72 (SD = 3.15). For the statement “to induce or enhance a spiritual experience”, red kratom users ranked it highest (position 6.70, SD = 3.15), while green strain users ranked it lowest (position 7.40, SD = 3.13). White kratom users ranked the statement at position 6.93 (SD = 3.24). For the statement “to be more sociable or enjoy social events more”, white kratom users ranked it highest (position 6.50, SD = 2.72), while red strain users ranked it lowest (position 7.26, SD = 2.46). Green kratom users ranked the statement at position 6.39 (SD = 2.54).

Moreover, for white strain users, there was a significant positive correlation (Figure 2) between the amount of kratom consumed per dose and the position at which the statement “to help you concentrate, work or study” was ranked, i.e., white kratom users who consumed higher doses tended to rank this statement higher. In contrast, the ranking position of the statement “to help you relax or sleep” was inversely correlated with the amount of kratom consumed per dose for white kratom users. These were the only significant correlations found between dosing amount and average ranking position across all the motivational statements.

### 3.4. Self-Reported Subjective Effects of Different Kratom Strains

Table 5 presents the self-reported effects of different kratom strains (green, red, and white) on various physiological outcomes and aspects of mood and cognition. Participants were asked to indicate on a visual slider (0–100) the extent to which they experienced a given drug effect when consuming a particular kratom strain. For every drug effect, a cross-strain comparison was made by means of an ANOVA. Out of the 39 VAS items, only six showed statistical significance in their cross-strain omnibus ANOVA, namely ‘being better able to concentrate’ (*p* < 0.01), ‘feeling more energetic’ (*p* < 0.01), ‘feeling more stimulated’ (*p* < 0.01), ‘feeling more constipated than usual’ (*p* < 0.01), ‘being better able to stay up all night’ (*p* < 0.01), ‘being better able to fall asleep’ (*p* = 0.0364). 

Tukey’s multiple comparison tests revealed significant differences between kratom strains for the following drug effects: ‘being better able to concentrate’, with green strains ($\bar{x}$ = 60.98) being rated to improve concentration more than red strains ($\bar{x}$ = 49.79, *p* < 0.01), and white strains ($\bar{x}$ = 69.42) being rated to improve concentration more than red strains (*p* < 0.01) and green strains (*p* = 0.04); ‘feeling more energetic’, with green strains ($\bar{x}$ = 64.46) being rated as more energizing than red strains ($\bar{x}$ = 52.93, *p* < 0.01), and white strains ($\bar{x}$ = 66.5) being rated as more energizing than red strains (*p* < 0.01); ‘feeling more stimulated’, with green strains ($\bar{x}$ = 52.94) being rated as less stimulating than white strains ($\bar{x}$ = 62.61, *p* = 0.01), and with red strains ($\bar{x}$ = 48.79) being rated as less stimulating than white strains (*p* < 0.01); ‘feeling more constipated than usually’, with green strains ($\bar{x}$ = 26.45) being rated as causing less constipation than red strains ($\bar{x}$ = 39.39, *p* < 0.01) and white strains ($\bar{x}$ = 41.74, *p* = 0.01); ‘being better able to stay up all night’, with green strains ($\bar{x}$ = 26.45) being rated as less effective in maintaining wakefulness compared to white strains ($\bar{x}$ = 41.14, *p* < 0.01), and with red strains ($\bar{x}$ = 18.43) being rated as less effective in maintaining wakefulness compared to white strains (*p* < 0.01).

### 3.5. Certificates of Analyses (COAs)

Analysis of Alkaloid Content of Super Speciosa Products was conducted by Santé Laboratories, which operate independently from Super Speciosa. The certificates of analyses (COAs) showing the alkaloid content of the products of Super Speciosa were obtained by means of liquid chromatography quadrupole Time-of-Flight Mass Spectrometry (LC-MS-QTOF). As shown in Table 6, the presence of mitragynine, paynantheine, speciogynine and speciocilliatine was measured for each product, as well as the total alkaloid content. The examined products were White Maeng Da, Red Maeng Da, Green Bali, White Thai, Green Maeng Da and Red Bali. A one-way ANOVA revealed that the products did not differ significantly from one another in terms of their mitragynine (*p* = 0.362), paynantheine (*p* = 0.917), speciogynine (0.803), or speciociliatine content (*p* = 0.762), and there was no significant difference in total alkaloid content (0.500).

## 4. Discussion

Kratom products are sold in the West as different strains, often denoted by the coloring of the plant product and the region where the plant was cultivated. Anecdotal reports of kratom users as well as the marketing of kratom products, both suggest that the effects induced by kratom are strain-dependent, presumably due to variations in the alkaloid content of different strains [25]. Given the absence of published research investigating the differences between kratom strains, the current study sought to investigate by means of an online questionnaire whether different color strains of kratom can induce distinct pharmacological effects in humans, and whether the use of different color strains is driven by distinct motivations.

The survey population can be described as primarily working middle-aged, Caucasian, American individuals with some level of post-secondary education. Given that most respondents were customers of an American-based online vendor, it is unsurprising that the survey population is predominantly American. However, the lack of ethnic diversity in the sample should be considered when interpreting the results of this study.

In the current survey, motivations for using either red, green, or white kratom strains were investigated by asking survey respondents to rank 13 pre-written motivational statements in the order most applicable to their use of the respective strains. Interestingly, there are notable congruencies between the respondents’ ranking of these statements and the way that different color strains are commonly marketed. For example, red kratom users ranking the motivation “to help you relax or sleep” at the highest position is consistent with product descriptions claiming that red strains are calming, anxiolytic and beneficial in treating insomnia. Likewise, the motivation “to improve your mood or to feel less sadness/depression” being ranked highest by green kratom users, is in line with marketing claims that green strains are the best for promoting overall well-being. As for white strain users, the finding that “to help you concentrate on work or study” was ranked highest, is congruent with descriptions of white strains being nootropic, stimulating and energizing. Interestingly, part of these congruencies extended to the results of the VAS scales, on which respondents indicated to what extent 39 drug effects were present when consuming red, green, or white kratom strains. Although all measures in the current study were self-report measures subject to reporting biases, there is a clear conceptual difference between self-reported motivations to use different kratom strains and self-reported effects of different kratom strains. Namely, the former represents a construct relating to motives and intentions whereas the latter represents actual experiences, albeit subjective and retrospectively observed.

It is therefore interesting that there was a strain hierarchy present in the results of VAS items pertaining to concentration, wakefulness, sleepiness, stimulation, and energy. Namely, the results consistently indicated that white strains were experienced to be the most stimulating and energizing, while green strains were experienced to be less stimulating and energizing than white strains but more stimulating and energizing than red strains. It can be argued that this hierarchy closely matches the aforementioned marketing claims and anecdotal reports of users consuming different kratom strains.

Interestingly however, the COAs of the different kratom products did not show any significant differences in alkaloid content between the kratom strains consumed by the survey population. Therefore, it seems that the significant between-strain differences in self-reported effects are unlikely to be explained by variations in alkaloid content. It is possible that the observed differences were caused by a placebo or expectancy effect induced by marketing information and/or by hearsay. If this were to be true, then the observed effect differences would not have a tangible pharmacological origin. However, an alternative explanation is that the observed differences are pharmacologically ‘real’, but that they are explained either by variations in alkaloids that were not measured or by other substituents of kratom, e.g., terpenes [25]. Yet, another possible explanation is differential metabolism and thus exposure of kratom alkaloids and other constituents as a function of interindividual metabolic differences. This has not been evaluated to date, but it is known that several kratom alkaloids may induce or inhibit metabolic enzymes [2,26].

It is important to note that most respondents of this survey were clientele of the same online kratom vendor, which can be seen both as a limitation and a strength of this study. Namely, the homogeneity of the sample likely impairs generalizability of the current results to other vendors or other kratom products. Moreover, with the survey population being comprised predominantly of working middle-aged, Caucasian, American individuals with some level of post-secondary education, the survey population was lacking in ethnic and national diversity. On the other hand, having a homogeneous survey population presumably increased the statistical power of this study compared to a (hypothetical) study in which the same number of respondents are recruited from different vendors. It is noteworthy that although more than one-third of the survey population was diagnosed with chronic pain, only 7.1% percent of respondents reported taking opioid pain medication. This adds credence to the claim that kratom has the potential to be a substitute for traditional analgesics, which is a notion that is increasingly being investigated by scientific and governmental institutions [8,27].

## 5. Conclusions

The current study found that despite a lack of detectable differences in alkaloid content across red, green, and white kratom strains, kratom users reported distinct subjective experiences associated with each strain, and these experiences mirrored the strains’ respective marketing descriptions, suggesting a potential influence of user expectations and marketing claims on the individual’s experience of different kratom strains. To assess the generalizability of these results, future survey studies should target more diverse populations while covering kratom strains from different vendors. One important component of kratom is its varied phytochemical composition which has so far primarily focused on predominant alkaloids. However, it cannot be excluded that minor alkaloids or non-alkaloid substances may account for the experienced differences between strains, which is an alternative explanation aside from user expectation. Controlled, double-blind clinical studies comparing the effects of well characterized kratom strains could experimentally prove whether the reported effect differences are caused by a placebo effect or by other factors, such as variations in terpenes or minor alkaloids that are not commonly quantified.

## Figures and Tables

**Figure 1 ijerph-20-06425-f001:**
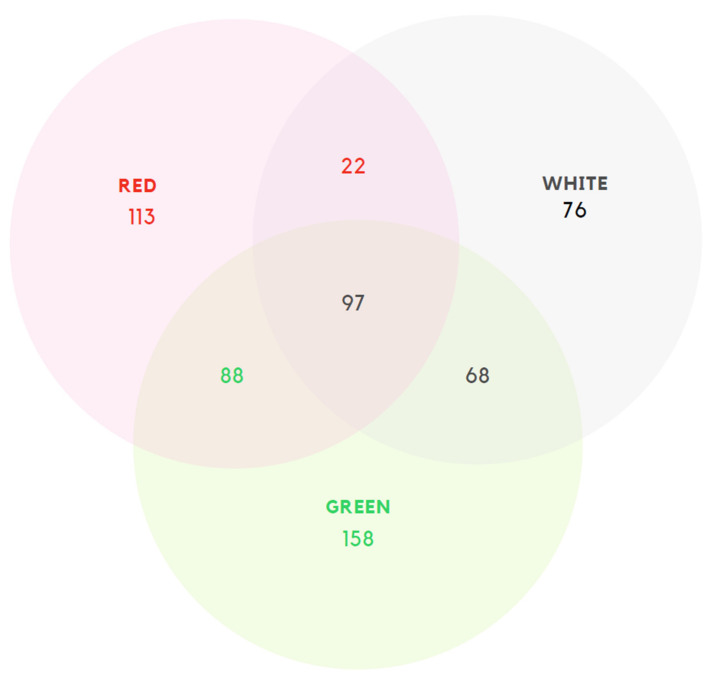
Number of users of kratom “strains” alone or in combination.

**Figure 2 ijerph-20-06425-f002:**
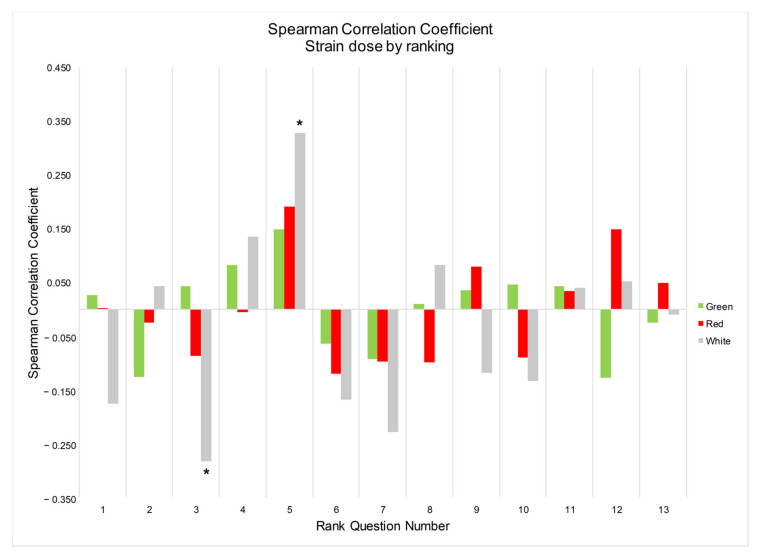
Spearman correlation coefficients between the ranking position of the motivational statements and the amount of kratom consumed per dose. See numbers in Table 4 for respective rank question matched to number. * *p* < 0.05 for significant correlation between dose and ranking. Positive correlation coefficients indicate higher amount per dose and higher ranking. Negative correlation coefficients indicate lower amount per dose and higher ranking. E.g., for white kratom use, lower dosing was associated with higher ranking of the statement “to help you relax or sleep”, i.e., white kratom users who used lower doses ranked this statement higher on average.

**Table 1 ijerph-20-06425-t001:** Representative effect descriptions of red, green, and white kratom strains. The text quotes in this table were taken from websites listed on the GMP-qualified vendors list of the American Kratom Association (https://www.americankratom.org/, accessed on 17 July 2023).

Vendor	Red	Green	White	URL (All Accessed on 17 July 2023)
Whole Herbs Kratom	“Red Kratom strains are more potent and have a soothing effect.”	“Green Kratom strains are more used to **calm** and **reduce pain**.”	“White strains can better serve as a **stimulant** to activate and boost the immune system.”	https://bit.ly/45ilXAl
Christopher’sOrganic Botanicals	“Reds are claimed to be more for **nighttime**, or late in the day in general as reported by consumers.”	“Green Kratom is said to be more for **daytime** use, or when more energy is required.”	“White kratom products are used **during the day** for increased **energy** and **focus**.”	Red: https://bit.ly/45jlmyp, Green: https://bit.ly/3WpvpOj, White: https://bit.ly/3q7bMOG
Super Speciosa	“Reds are for **relaxation**.”	“Greens are for **energy**.”	“Whites are viewed as hybrids for both **energy** and **focus**.”	https://superspeciosa.com/new-to-kratom/
PurKratom	“Red vein kratom is commonly used when **rest** is needed or at **bedtime**.”	“Green vein kratom is often used **during the day**.”	“White vein kratom is typically consumed **in the morning**.”	https://www.purkratom.com/kratom-strains-101-learn-about-the-different-strains/
Salvia Hut	“**Sedative** effect which allows for the user to be **calm** and acts as an **analgesic**. Simultaneously, it is also known to be a great aid for individuals with **insomnia**.”	“**Not as mellow as red vein** Kratom and simultaneously, it is **not as potent as white vein** Kratom.”	“It is considered to have a **stimulant** effect. Typically, white vein Kratom is used as a **replacement for coffee** as its base effects are **energy** boosts and a sense of **alertness**.”	https://salviahut.com/types-of-kratom-strains-and-their-effects/
Nuwave Botanicals	“Best for **rest** and **relaxing**.”	“Best for **balance**, **confidence**, and **inspiration**.”	“Best kratom for **energy**.”	https://soapkorner.com/a-brief-introduction-to-kratom-strains/
Buy Kratom Bulk USA	“The longer drying time and/or fermentation of red vein kratom generally enhances the alkaloids associated with **relaxation** over energy.”	“Green vein kratom maintains a greater balance of alkaloids found in both red vein kratom and white vein kratom due to its process that falls in between the two.”	“Sometimes, ground stems from kratom leaves are added to white strains to add more of the **stimulating** alkaloids that are naturally found in kratom veins.”	https://nuwavebotanicals.com/what-is-kratom-powder/

**Table 2 ijerph-20-06425-t002:** General demographic information of survey respondents including alcohol and nicotine product consumption.

	Frequency	Percent
**Age**		
18–24	9	2.2%
25–34	81	20.0%
35–44	120	29.7%
45–54	92	22.8%
55–64	74	18.3%
65 or older	28	6.9%
**Gender**		
Male	227	56.6%
Female	165	41.1%
Non-Binary	6	1.5%
Prefer not to say	3	0.7%
**Biological Sex**		
Male	229	57.0%
Female	168	41.8%
Other/Prefer not to say	5	1.2%
**Education**		
Did not complete high school	13	3.2%
High school graduate or equivalent	74	18.3%
Some college (e.g., AA, AS, or no degree)	162	40.1%
Prefer not to say	5	1.2%
Doctorate	8	2.0%
Bachelor’s degree (e.g., BA, BS, BSc, AB)	100	24.8%
Advanced Degree (e.g., MA, MS, MSc, MBA, PhD, MD)	42	10.4%
**Employment Status**		
Employed for wages	258	64.0%
Employed-currently off sick	2	0.5%
Out of work for less than a year	11	2.7%
Out of work for 1 year or longer	7	1.7%
Homeworker	25	6.2%
Student	2	0.5%
Unable to work	36	8.9%
Retired	46	11.4%
Prefer not to say	16	4.0%
**Marital Status**		
Married	179	44.5%
Widowed	10	2.5%
Divorced	77	19.2%
Separated	7	1.7%
Never married	129	32.1%
**Nationality**		
American	276	85.98%
Native American	7	2.18%
German	4	1.25%
Irish	9	2.80%
Italian	5	1.56%
Other	20	6.23%
**Ethnicity**		
Caucasian	319	86.4%
Hispanic	18	4.9%
Black	4	1.08%
Asian	2	0.54%
Mixed	26	7.05%
**Annual Household Income**		
Less than $10,000	19	4.7%
$10,000–$19,999	34	8.5%
$20,000–$29,999	28	7.0%
$30,000–$39,999	34	8.5%
$40,000–$49,999	34	8.5%
$50,000–$59,999	31	7.7%
$60,000–$69,999	24	6.0%
$70,000–$79,999	26	6.5%
$80,000–$89,999	13	3.2%
$90,000–$99,999	22	5.5%
$100,000–$149,999	64	16.0%
More than $150,000	42	10.5%
Prefer not to say	30	7.5%
**How often do you smoke/vape?**		
Never or rarely	404	62.7%
Daily	211	32.8%
At least once a week	11	1.7%
Several times a week	18	2.8%
**How often do you consume alcohol?**		
Never or rarely	490	76.1%
Daily	25	3.9%
At least once a week	87	13.5%
Several times a week	42	6.5%

**Table 3 ijerph-20-06425-t003:** Clinical Profile and Medical History of the Survey Population.

	Frequency	Percent
**Prescription Medications (choose all that apply)**		
Antidepressants (SSRI’s, tricyclic antidepressants)	187	29.0%
Anxiety medication (benzodiazepines, e.g., Xanax)	92	14.3%
Antipsychotics (e.g., quetiapine, olanzapine, risperidone)	21	3.3%
Opioid pain killers (e.g., fentanyl, morphine, codeine)	46	7.1%
Stimulants (e.g., Ritalin, amphetamines, etc.)	60	9.3%
Other medications	153	23.8%
No medications	287	44.6%
**Clinical Diagnoses (choose all that apply)**		
Post-traumatic Stress Disorder (PTSD)	134	20.8%
Depression (major depressive disorder/persistent depressive disorder, dysthymia)	275	42.7%
Social Anxiety Disorder	115	17.9%
Generalized Anxiety Disorder	211	32.8%
Schizophrenia	10	1.6%
Bipolar Disorder	56	8.7%
Personality Disorder	19	3.0%
Attention Deficit Disorder (ADD)/Attention Deficit Hyperactivity Disorder (ADHD)	128	19.9%
Addiction/substance use disorder	92	14.3%
Fibromyalgia	56	8.7%
Rheumatoid Arthritis	51	7.9%
Chronic Pain	250	38.8%
None of the above diagnoses	142	22.0%

**Table 4 ijerph-20-06425-t004:** Between-Strain Comparison of Motivations. Survey respondents were shown 13 motivational statements in random order and were asked to rank the statements in the order most applicable to their use of the given kratom strain (red, green, or white). The χ^2^ statistic signifies between-strain differences in the average ranking position of the respective motivation. For motivations that had significant between-strain differences in ranking position according to the χ^2^ statistic, the ‘Rank’ column indicates the categorical ranking on a scale from 1 to 3 (based on the average ranking of the respective motivation), with 1 being the highest and 3 being the lowest. E.g., users ranked red strains as most beneficial to “treat a medical condition” while green strains were perceived as least beneficial. Overall, use for this condition ranked second highest among all 13 motivation statements.

		Green (N = 288)			Red (N = 184)			White (N = 131)				
Question	Motivation Statement	Mean	Std	Rank	Mean	Std	Rank	Mean	Std	Rank	χ^2^ Statistic	*p*-Value
1	To feel less anxiety and/or stress	2.83	1.73	NS	3.03	1.72	NS	3.20	2.02	NS	3.542	0.170
2	To treat a medical condition	4.25	3.51	3	3.51	3.37	1	4.18	3.38	2	6.029	0.049
3	To help you relax or sleep	5.15	2.68	2	3.83	2.54	1	6.06	2.99	3	25.389	<0.001
4	To improve your mood or to feel less sadness/depression	5.28	4.00	1	6.34	3.88	3	5.45	3.96	2	9.816	0.007
5	To help you concentrate, work or study	5.72	3.15	2	6.97	3.15	3	5.30	3.49	1	14.776	0.001
6	To feel elated, euphoric or intoxicated	6.73	2.90	NS	6.56	2.96	NS	6.62	3.03	NS	2.299	0.317
7	To induce or enhance a spiritual experience	7.40	3.13	3	6.70	3.15	1	6.93	3.24	2	7.726	0.021
8	To be more sociable or to get more enjoyment out of social events	6.39	2.54	2	7.26	2.46	3	6.50	2.72	1	10.424	0.005
9	To stay awake longer or to prolong a night out with friends	8.10	2.49	NS	8.43	2.06	NS	7.54	2.38	NS	3.902	0.142
10	To improve the quality of sex	8.90	2.32	NS	8.61	2.45	NS	8.60	2.45	NS	3.918	0.141
11	To lose weight or to reduce appetite	8.90	2.30	NS	8.64	2.29	NS	8.76	2.38	NS	0.196	0.907
12	To improve the effects of other substances	10.53	2.15	NS	10.44	2.22	NS	10.75	1.99	NS	0.675	0.714
13	Other (please specify)	10.82	4.32	NS	10.66	4.50	NS	11.11	4.04	NS	2.571	0.276

**Table 5 ijerph-20-06425-t005:** Self-reported effects of different kratom strains. Participants were instructed to drag a visual slider between 0 and 100 to indicate the extent to which they experienced a given drug effect when consuming green, red, or white kratom. Only the drug effect that presented with significant F-test statistic was further analyzed using post-hoc test to determine the difference between the strains for that effect.

	Green			Red			White			F-Test
VAS Drug Effect	N	Mean	Std	N	Mean	Std	N	Mean	Std	*p*-Value
Feeling less physical pain	254	71.75	26.85	168	74.68	25.96	108	67.03	25.82	0.0637
Feeling happier	256	68.16	26.34	144	65.38	28.79	118	67.91	24.62	0.5850
Feeling more content	236	67.22	25.82	148	63.18	29.03	106	68.61	23.71	0.2066
Feeling more relaxed	248	63.82	26.08	152	65.51	26.66	94	60.23	24.32	0.2964
Feeling more nervous/tense	82	12.24	20.63	46	5.33	11.46	32	10.38	16.44	0.1055
Feeling calmer	230	64.55	25.74	156	65.24	27.38	105	58.71	27.6	0.1115
Feeling more on edge	91	14.09	23.46	44	8.23	17.64	35	8.91	11.58	0.1994
Being more easily agitated	87	18.72	24.67	57	12.98	23.45	30	12.87	15.72	0.2556
Having more mood swings	83	18.33	24.95	52	11.04	16.88	29	12.86	15.9	0.1295
Being better able to concentrate	204	60.98 *^,#^	26.91	121	49.79 *^,$^	30.57	95	69.42 ^#,$^	24.31	*p* < 0.01, * < 0.01, green vs. red # 0.04, green vs. white $ < 0.01, red vs. white
Being more easily distracted	91	17.10	28.44	54	13.70	21.81	38	21.74	22.87	0.3331
Feeling less depressed	232	66.50	28.56	136	64.86	27.25	101	68.34	24.16	0.6233
Feeling more anxious	81	12.67	21.46	43	8.33	15.97	30	11.57	18.35	0.4972
Being more forgetful	94	19.94	27.50	55	16.27	22.67	31	19.19	27.46	0.7055
Being less forgetful	121	38.59	32.05	74	33.09	29.48	60	46.07	31.55	0.0586
Feeling more energetic	251	64.46 *	26.12	116	52.93 *^,#^	31.60	112	66.5 ^#^	27.11	*p* < 0.01, * < 0.01, green vs. red # < 0.01, red vs. white
Feeling more fatigued	88	15.85	24.40	64	21.38	26.47	31	21.48	29.85	0.3565
Feeling more stimulated	220	52.94 *	27.35	95	48.79 ^#^	31.91	100	62.61 *^,#^	26.9	*p* < 0.01, * 0.01, green vs. white # < 0.01, red vs. white
Feeling more nauseous	104	19.52	21.11	70	20.01	23.92	35	16.31	16.05	0.6836
Feeling more constipated than usually	149	26.45 *^,#^	32.40	93	39.39 *	33.94	53	41.74 ^#^	33.44	*p* < 0.01, * < 0.01, green vs. red # 0.01, green vs. white
Vomiting more than usually	64	8.48	14.49	47	14.23	24.46	25	6.28	9.11	0.1295
Enjoying social events more than usually	208	61.45	27.40	116	55.86	32.98	91	63.65	27.1	0.1203
Being better able to stay up all night	102	26.45 *	28.09	54	18.43 ^#^	25.47	58	41.14 *^,#^	31.73	*p* < 0.01, * < 0.01, green vs. white # < 0.01, red vs. white
Being better able to fall asleep	195	54.56 *	33.12	141	62.73 *	31.42	69	53	30.56	*p* = 0.0364, * 0.0566, green vs. red
Having less insomnia	158	48.86	34.06	119	57.95	33.54	55	52.07	30.8	0.0812
Having more insomnia	76	18.00	26.68	48	12.17	20.85	24	22.08	24.86	0.2291
Feeling less sociable than usually	80	16.84	29.20	52	11.38	18.63	28	10.07	18.93	0.3079
Having more diarrhea than usually	64	4.97	15.87	53	5.47	12.32	25	7.92	18.78	0.7087
Having more stomachache than usually	91	17.74	23.97	61	19.67	24.70	33	15.15	20.15	0.6723
Feeling less withdrawal symptoms when withdrawing from other opioids (e.g., heroin, fentanyl etc.)	113	52.26	41.94	78	51.55	40.61	52	48.69	41.19	0.8735
Feeling less withdrawal symptoms when withdrawing from other substances that are not opioids (e.g., MDMA, cocaine, amphetamine, LSD, psilocybin)	93	32.86	39.29	73	41.37	40.19	41	43.8	37.32	0.2222
Feeling more socially withdrawn	82	10.21	20.99	52	6.65	16.35	26	8.46	20.14	0.5860
Feeling more sociable	211	62.05	28.54	123	56.85	30.44	100	61.09	29	0.2808
Not worrying as much	229	59.71	26.18	137	56.06	30.01	103	58.31	28.43	0.4792
Being less bothered by the actions of others	205	53.30	28.64	122	54.50	26.42	88	55.34	26.88	0.8293
Having an increased libido	104	33.25	28.85	67	26.81	31.76	44	34.02	29.44	0.2528
Having a decreased libido	90	21.81	28.85	55	24.04	31.93	44	30.61	33.91	0.3025
Experiencing more sexual satisfaction	107	35.11	31.88	60	33.45	32.87	42	31.69	29.9	0.8314
Having greater sexual dysfunction	82	22.59	30.87	50	18.04	27.28	37	23.51	30.24	0.6206

**Table 6 ijerph-20-06425-t006:** Certificates of Analysis (COAs) for Super Speciosa kratom products from three different batches. Santé Laboratories provided COAs of the alkaloid content for the included kratom products of Super Speciosa. Santé Laboratories operate independently and are unaffiliated with Super Speciosa.

		White Maeng Da	Red Maeng Da	Green Bali	White Thai	Green Maeng Da	Red Bali	F-Statistic	*p*-Value
Mitragynine	Mean	1.54%	1.52%	1.44%	1.41%	1.56%	1.4%	1.21	0.362
	%CV	6.2	6.42	9.12	8.84	7.04	7.3		
Paynantheine	Mean	0.28%	0.3%	0.28%	0.27%	0.29%	0.27%	0.278	0.917
	%CV	0.43	0.41	0.39	0.56	0.48	0.06		
Speciogynine	Mean	0.22%	0.23%	0.22%	0.21%	0.22%	0.21%	0.453	0.803
	%CV	0.16	0.21	0.17	0.22	0.17	0.02		
Speciociliatine	Mean	0.4%	0.32%	0.35%	0.36%	0.35%	0.32%	0.513	0.762
	%CV	1.16	0.82	2.22	1.36	0.57	3.08		
Total Alkaloid Content	Mean	2.44%	2.37%	2.28%	2.25%	2.42%	2.2%	0.92	0.5
	%CV	0.75	1.04	1.98	1.72	0.73	1.94		

## Data Availability

The data presented in this study are available on request from the corresponding author. The data are not publicly available due to privacy of participating respondents.

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
