# Peer review of "Examining the Psychoactive Differences between Kratom Strains"

_ijerph, 2023, doi:10.3390/ijerph20146425_

Round 1
Reviewer 1 Report
Overall the content is okay and meets the expectation. Kratom is a quite famous plant because of the ability of psychoactive properties.
(1) The author needs to find good keywords. Suggestion: Kratom, Mitragyna speciosa, Psychoactive compound, Mitragynine, 7-hydroxymitragynine.
(2) Tables 2 and 5 needs to improve - Table format and data representation.
(3) Data representation needs to improve.
(4) Please re-write the comprehensive conclusion.
(5) Most of the survey and collecting data usually involve money and expenses. The author state that this study does not involve any funding. Please check this.
(6) The ethical committee approval should also mention in section 2.2.
The quality of the English language can be improved.
Author Response
Overall the content is okay and meets the expectation. Kratom is a quite famous plant because of the ability of psychoactive properties.
(1) The author needs to find good keywords. Suggestion: Kratom, Mitragyna speciosa, Psychoactive compound, Mitragynine, 7-hydroxymitragynine.
Response: Thank you and the new keywords have now been provided.
(2) Tables 2 and 5 needs to improve - Table format and data representation.
Response: we have attempted to improve the tables while also keeping the IJERPH required formatting in mind. By reducing the font size the tables are not as large as they used to be. However, the data displayed is important in regards to demographics and including the necessary information to determine perceived differences between strains by respondents. Although we wanted to be transparent by including all the test statistics, we now removed the F-test results column in table 5 and added the comparisons behind the p-values.
(3) Data representation needs to improve.
Response: we now have included further clarification with each table to better interpret it. We hope that some of the examples we provided will improve ease of interpretation and connecting the tables with the results and discussion sections.
(4) Please re-write the comprehensive conclusion.
Response: we have expanded on the conclusions in regards to the alignment of perceived effects with marketing but also to account in more detail for factors that we could not evaluate in this study, especially the contributions of other substances present in kratom leaf material.
(5) Most of the survey and collecting data usually involve money and expenses. The author state that this study does not involve any funding. Please check this.
Response: we indeed did not receive any funding for this survey or the data analysis. The work was conducted using an institutional license of Qualtrics freely available to the investigative team. Statistical software was also available through the institutions. No participants were compensated for completing the survey.
(6) The ethical committee approval should also mention in section 2.2.
Response: we have added the following sentence at the end of section 2.2: “Ethical approval was obtained from the Ethics Review Committee of Psychology and Neuroscience of Maastricht University (ERCPN-226_101_08_2020_A2).”
Reviewer 2 Report
The present study aims to investigate kratom products marketed as red, green and white kratom strains and their relationship to unique pharmacological effects in humans.
The work has important methodological and knowledge gaps, e.g..
1. In the introduction:
a. It does not cite phytochemical studies denoting the presence of plant components in different parts of the plant.
b. Does not specify which part of the plant is consumed or is more related to the mentioned effects, remember that metabolites are concentrated in different concentrations in different parts of the plant.
c. Does not consider what type of ecosystem predominates where the plant is grown to understand its potential (concentration of metabolites).
d. It does not consider that the effects are the product of pharmacological interactions between the metabolites they contain and that the diversity of effects is related to the abundance of these, not to a single component in particular.
2. On methodology:
a. Despite being a study involving a survey and not an intervention in humans, it should be observed by an ethics committee since it is information that can be traced for clinical purposes.
b. It does not present a letter of informed consent to use the answers provided in the survey.
c. The instrument does not present validation indicating the impartiality and non-bias of the questions included.
d. It should include the abundance of the main metabolites of kraptom with the effect reported in the literature and the one commented by the study subjects.
e. Does not consider in the study design the effect of the drug in toto.
3. In discussion and conclusion:
Concludes that despite the lack of detectable differences in alkaloid content between the red, green, and white kratom strains, users reported distinct subjective experiences associated with each strain, and these experiences mirrored the respective marketing descriptions of the strains, suggesting a strong influence of user expectations and marketing claims on individual experience of the different kratom strains.
The conclusion does not provide information of value regarding the relationship of components, use and observed clinical effect, but merely states marketing claims.
It does not meet the stated objective, as it does not provide information of value with respect to marketed kratom products in relation to unique pharmacological effects in humans, despite knowing the metabolite profile.
Author Response
The present study aims to investigate kratom products marketed as red, green and white kratom strains and their relationship to unique pharmacological effects in humans.
The work has important methodological and knowledge gaps, e.g..
- In the introduction:
- It does not cite phytochemical studies denoting the presence of plant components in different parts of the plant.
Response: Although there have been reports of the phytochemical analysis of other parts of the plant, only the leaves are used for medicinal and recreational purposes. We have included a sentence with reference in the introduction: “Although other parts of the tree have been analyzed for their phytochemical composition, only the leaves of Mitragyna speciosa are used for medicinal and recreational purposes.[6]”
- Does not specify which part of the plant is consumed or is more related to the mentioned effects, remember that metabolites are concentrated in different concentrations in different parts of the plant.
Response: we have now emphasized in the beginning of the introduction that the leaves are the part of the plant that are used and sold. In regards to metabolites, does this refer to secondary plant metabolites? If so, then the alkaloids have been the primary (and only) target of pharmacological investigation ever since mitragynine was isolated back in 1923 in England.
- Does not consider what type of ecosystem predominates where the plant is grown to understand its potential (concentration of metabolites).
Response: we have now expanded substantially on this topic in the introduction: “One phytochemical study did examine the metabolomic profile of young and mature kratom leaves and identified 5 unique metabolites between the young and mature leaves while 76 secondary metabolites were present in both leaf samples, albeit in dif-ferent concentrations.[13] While the major alkaloid mitragynine is present in approx-imately equal amounts in both young and mature leaves, a number of other indole alkaloids, such as speciogynine, speciociliatine, and 7-hyroxymitragynine are present in higher abundance in mature leaves. In most kratom leaf materials available on the US market, mature leaves are used. Another study examined the seasonal and geo-graphical differences between red- and green-veined kratom in parts of Thailand.[14] Though mitragynine remained the most abundant alkaloid, the total alkaloid concen-tration was much lower in the fall (October) sample compared to winter (January) and summer (June) samples. Regional differences indicate that total alkoid content does also vary.”
- It does not consider that the effects are the product of pharmacological interactions between the metabolites they contain and that the diversity of effects is related to the abundance of these, not to a single component in particular.
Response: we do recognize that our analysis is limited in scope given that we only analyzed three batches of 6 different products from one vendor. However, we do believe that the differences between strains, given the information available, is well explored in this manuscript. We state in the discussion and the conclusions that more research is needed to explore whether specific compounds/metabolites present in kratom may contribute to distinct effects. But based on the Certificates of Analysis it is clear that the major alkaloids present in the leaf material did not differ.
- On methodology:
- Despite being a study involving a survey and not an intervention in humans, it should be observed by an ethics committee since it is information that can be traced for clinical purposes.
Response: this was mentioned at the end of the survey. We did add the ethical approval in section 2.2 as follows: “Ethical approval was obtained from the Ethics Review Committee of Psychology and Neuroscience of Maastricht University (ERCPN-226_101_08_2020_A2).”
- It does not present a letter of informed consent to use the answers provided in the survey.
Response: the entire survey is attached in appendix 1. The survey has an informed consent at the beginning. We included the following sentence in section 2.2: “Informed consent was obtained at the beginning of the survey from all participants prior to starting the survey (see appendix 1).”
- The instrument does not present validation indicating the impartiality and non-bias of the questions included.
Response: several questions used in this survey were taken from prior kratom surveys that have been successfully reviewed and published. The ranking of motivations for use was published before as were the use of visual analogue scales (VAS) that are commonly used in surveys. We refer to the respective references included in the method subsections describing the questionnaire design.
- It should include the abundance of the main metabolites of kraptom with the effect reported in the literature and the one commented by the study subjects.
Response: we have now included a brief summary of the known effects of kratom alkaloids that have been reported in the literature in recent years. Of note, no kratom alkaloid or isolated metabolite has been tested in clinical trials to date in regards to its specific effects. We are therefore reluctant to link effects reported in animal models to those expressed by participants in this study.
- Does not consider in the study design the effect of the drug in toto.
Response: we are uncertain what this means. Kratom was used as an available commercial product by users. The analysis of alkaloids provided in this study is not meant to be a total representation of the composition of kratom. We also clarify that based on our current knowledge and limited insights into the composition of commercial kratom products, we cannot conclude whether marketing or the chemical composition of kratom products can explain any perceived subjective differences between strains.
- In discussion and conclusion:
Concludes that despite the lack of detectable differences in alkaloid content between the red, green, and white kratom strains, users reported distinct subjective experiences associated with each strain, and these experiences mirrored the respective marketing descriptions of the strains, suggesting a strong influence of user expectations and marketing claims on individual experience of the different kratom strains.
The conclusion does not provide information of value regarding the relationship of components, use and observed clinical effect, but merely states marketing claims.
Response: because all investigations into kratom effects to date are based on pre-clinical animal and in vitro experiments, and the fact that commercial kratom products contain variable amounts of alkaloids and other substances, we cannot reach any conclusion about specific components in the kratom products. We further would like to point out that in the traditional use setting, i.e., in Southeast Asia, there is no distinction made between different colored kratom strains.
It does not meet the stated objective, as it does not provide information of value with respect to marketed kratom products in relation to unique pharmacological effects in humans, despite knowing the metabolite profile.
Response: the survey explored distinct motivations for use which were aligned with kratom use motivations. Although we were not able to obtain a complete metabolomic profile of all components present in each of the evaluated kratom products, the major alkaloids included in the analysis across 3 batches provide sufficient insights into the variability in composition of these products. We do state that this work has limitations because only kratom products from one vendor were included in the analysis. Whether the self-perceived effect of kratom users is objectively associated with a distinct metabolite profile cannot be answered at this point. However, it is clear that kratom users do perceive at least some distinct differences between kratom strains, in the absence of a significant difference in metabolite profile based on the provided COAs.
Round 2
Reviewer 2 Report
The article has been substantially improved, however, it should be noted that the scope is merely qualitative and perceptual. And in the future it is necessary to support the associated effects with biological models.